# Innovative Use of Self-Attention-Based Ensemble Deep Learning for Suicide Risk Detection in Social Media Posts

**Hoan-Suk Choi** [1] and **Jinhong Yang** [2,*]

1   KAIST-Megazone Cloud Intelligent Cloud Computing Convergence Research Center,
    Daejeon 34141, Republic of Korea; choihs@mz.co.kr
2   Department of Medical IT, INJE University, Gimhae 50843, Republic of Korea
*   Correspondence: jinhong@inje.ac.kr; Tel.: +82-10-8518-4555

**Abstract:** Suicidal ideation constitutes a critical concern in mental health, adversely affecting individuals and society at large. The early detection of such ideation is vital for providing timely support to individuals and mitigating its societal impact. With social media serving as a platform for self-expression, it offers a rich source of data that can reveal early symptoms of mental health issues. This paper introduces an innovative ensemble learning method named LSTM-Attention-BiTCN, which fuses LSTM and BiTCN models with a self-attention mechanism to detect signs of suicidality in social media posts. Our LSTM-Attention-BiTCN model demonstrated superior performance in comparison to baseline models in the realm of classification and suicidal ideation detection, boasting an accuracy of 0.9405, a precision of 0.9385, a recall of 0.9424, and an F1-score of 0.9405. Our proposed model can aid healthcare professionals in recognizing suicidal tendencies among social media users accurately, thereby contributing to efforts to reduce suicide rates.

**Keywords:** bidirectional TCN; LSTM; NLP; self-attention; suicidal ideation





## 1. Introduction

Suicide constitutes a significant impediment within the realm of mental health, recognized for its severity and distinctiveness. The World Health Organization has reported that annually, upwards of 700,000 individuals grapple with suicide-related behaviors and ideation, impacting an even greater number on a psychological level [1]. In the context of the United States, the year 2021 witnessed approximately 1.7 million suicide attempts, out of which 48,183 were fatal [2], indicating that those engaging in self-harm far outnumber fatalities. This underscores the potential for intervention, as the onset of suicidal thoughts can be intercepted with timely recognition and response.

A surge of studies has delved into the multifaceted nature of suicidality, aiming to furnish healthcare practitioners with the requisite tools for its early detection. Research by Balbuena et al. [3] explores the long-term forecasters of suicide within clinical populations subjected to hospitalization, while Mbarek et al. [4] scrutinize social media platforms as a means to identify signs of suicidality through user profile analysis. Allen et al. [5] highlight the significance of immediate predictors of suicidal thoughts, such as emotional distress, social adversity, and disturbances in sleep. They advocate for the adoption of personal electronic devices and sophisticated computational methodologies to observe and interpret shifts in individual mood and behavioral patterns, thereby gauging suicide risk.

The ubiquity of social media in daily life renders it a rich repository of personal disclosures [6]. The transparency and interactive nature of these platforms encourage individuals to divulge personal experiences, including those of a sensitive or distressing nature [7]. Anonymity and the absence of physical barriers further contribute to their role as a monitor for mental state evaluations. In this context, Ji et al. [8] acknowledge the utility of social networks in offering valuable insights into emotional well-being and potential suicide risk, proposing that these platforms serve as ancillary channels for clinical assessment.

In this study, we introduce a hybrid prediction model that leverages advanced neural network architectures to differentiate between suicidal and non-suicidal narratives on social platforms. The cornerstone of our study is an ensemble model that integrates Long Short-Term Memory (LSTM), self-attention, and a bidirectional architecture of Temporal Convolutional Networks (LSTM-Attention-BiTCN). This ensemble model aims to discern expressions of suicidality within social media discourse. We employ a dataset from Reddit posts to reflect users' emotional states and cognitions, utilizing a fusion of state-of-the-art deep learning architectures.

The principal contributions of our study are as follows:

- The development of an LSTM-Attention-BiTCN ensemble model that accurately segregates suicidal ideation within social media postings from those that are non-suicidal.
- The utilization of social media datasets from social networks such as Reddit to mirror the emotional and cognitive states of individuals.
- The demonstration of the superior performance of our proposed model in identifying suicidal expressions, surpassing the predictive capabilities of several baseline models, including K-Nearest Neighbor, Random Forest, Decision Tree, Gradient Boost, XGBoost, Logistic Regression, LSTM, Bidirectional LSTM (BiLSTM), Convolutional Neural Networks (CNNs), and LSTM-CNN.

The rest of the manuscript is structured as follows: Section 2 presents an overview of the latest studies in identifying suicidal ideation. In Section 3, the proposed methodology is described in comprehensive detail. Section 4 showcases our experimental results for the designed model. Finally, Section 5 concludes our research and proposes avenues for future research in this domain.

## 2. Literature Review

The pervasive nature of social media has not only provided a platform for communication and social interaction but has also become a medium through which expressions of mental distress, including suicidality, are frequently manifested. This has led to a series of scholarly investigations aimed at harnessing social media's expansive reach to preemptively identify and mitigate the risks associated with suicide.

For example, Cao et al. [9] devised an advanced knowledge graph complemented by deep learning methodologies to monitor and assess the presence of suicidal expressions on digital platforms. This system employs a dual-layer attention mechanism, which highlights key predictors and explicit reasoning behind individuals' suicidal tendencies. Analyzing content from Reddit and other microblogging services, their approach pinpointed personality traits, user-generated content, and personal experiences as primary domains from a total of six, which could potentially signal suicidal ideations through analysis of text, imagery, emotional distress, and experiences of depression.

Another study by Ghosal et al. [10] incorporated fastText embedding coupled with an XGBoost classifier to sift through data effectively. They introduced a pioneering framework that segregates content that indicates suicidal thoughts from that which suggests depression. This model utilized the machine learning classifier XGBoost for refined categorization, the TF-IDF vector for highlighting term significance, and fastText embedding for nuanced contextual understanding.

Cusick et al. [11] explored a weakly supervised methodology to extract current indicators of suicidal ideation from unstructured notes within Electronic Health Records (EHRs). Leveraging a rule-based natural language processing approach, they annotated training and validation notes, which then served as a foundation for training Logistic Regression, Support Vector Machine, and CNN models. This approach, designed for comprehensive analysis of clinical documentation, has significant implications for concentrated suicide prevention strategies within clinical information systems.

Adarsh et al. [12] implemented a decision model that addresses the imbalanced representation across varying age groups and demographic backgrounds. They crafted an ensemble model integrating Support Vector Machine with K-Nearest Neighbor algorithms,

which possesses an inherent explainability and is complemented by mechanisms for correcting inaccuracies due to mislabeled data. This model demonstrated a notable classification accuracy rate of 98.05%, distinguishing between states of depression and suicidal ideation.

Tadesse et al. [13] conducted a thorough investigation into suicide ideation using a combination of n-gram analysis, an LSTM-CNN model, and additional machine learning techniques. They curated a dataset from the r/SuicideWatch subreddit featuring both suicidal and non-suicidal posts. Employing 10-fold cross-validation to refine the hyperparameters, their model integrates the analytical strengths of CNNs and LSTMs to process textual data. The model architecture utilizes an embedding layer for vector representation, a dropout layer to prevent overfitting, an LSTM layer for feature extraction, followed by a pooling layer, a flattening step, and a SoftMax function. Evaluation metrics, including F1 score, recall, accuracy, and precision, were applied to gauge the performance of the baseline model within this deep learning classification framework.

## 3. Methodology

In this research, we introduce an innovative model, denoted as LSTM-Attention-BiTCN, which is designed to enhance the precision and efficacy of identifying potential suicide ideation from user interactions on social networks. This model harnessed the textual data from Reddit, analyzing posts as reflections of the emotional and cognitive states of its users. It integrated state-of-the-art deep learning architectures, including LSTM and BiTCN, to dissect and comprehend the intricate patterns in the data. Our methodology encompasses multiple phases: data aggregation, preprocessing, embedding of words, model training, and an assessment of performance.

As illustrated in Figure 1, the initial step involved preprocessing Reddit posts by eliminating any extraneous elements that may noise the analysis. Subsequently, the cleaned input was fed into the embedding phase to distill and encapsulate the salient attributes of the textual data, thus refining the input for enhanced model accuracy. The LSTM-Attention-BiTCN classifier embarked on its analytic process by assigning the curated word sets to their specific vectors through the embedding layer. The LSTM component of the model was adept at unraveling and retaining word dependencies of considerable temporal extent. This LSTM-processed information was then channeled through a series of BiTCN layers designed to unravel extended temporal patterns embedded within the texts. An attention layer, following the BiTCN layers, served to selectively emphasize significant textual elements by assigning adaptive weights, thus sharpening the focus on crucial signals indicative of suicidal intent. After that, two pooling layers (average pooling and max pooling) processed the information separately, eventually merging their outputs to form a cohesive vector that entered the classification sequence. Finally, the classification process employed "ReLu" and "Sigmoid" activation functions within two dense layers to distill the probability of suicidal content within each post. To mitigate the risk of overfitting to the training dataset, two strategically placed dropout layers were integrated: one succeeding the embedding layer and another preceding the final dense layer.

### 3.1. Preprocessing

In the preprocessing stage, delineated in Figure 1, we applied filtering techniques to the collected social media data to extract and eliminate noise and irrelevant information that could impair the embedding model's capacity to learn effectively. During this stage, the raw textual data harvested from social networking platforms undergo a cleaning process. This included the elimination of punctuation, newline characters, URLs, hashtags, mentions, and similar non-content elements. To enhance the LSTM-Attention-BiTCN model's performance, we substituted instances of emails, URLs, mentions, and punctuation with a single whitespace. In addition to these steps, we standardized the text by converting it to lowercase and removing newline characters to ensure uniformity. The filtration process also involved the exclusion of stop words to maintain focus on the content's substantive elements. After purging the text of all undesired characters, we tokenized the posts using

the Natural Language Toolkit. This process broke down each sentence into a list of its tokens, facilitating further analysis. After tokenization, the cleaned dataset was ready for entry into the embedding layer. Here, it would be transformed into a structured vector format that the model can process to discern underlying emotional states and assess the risk of suicidality present in the text.

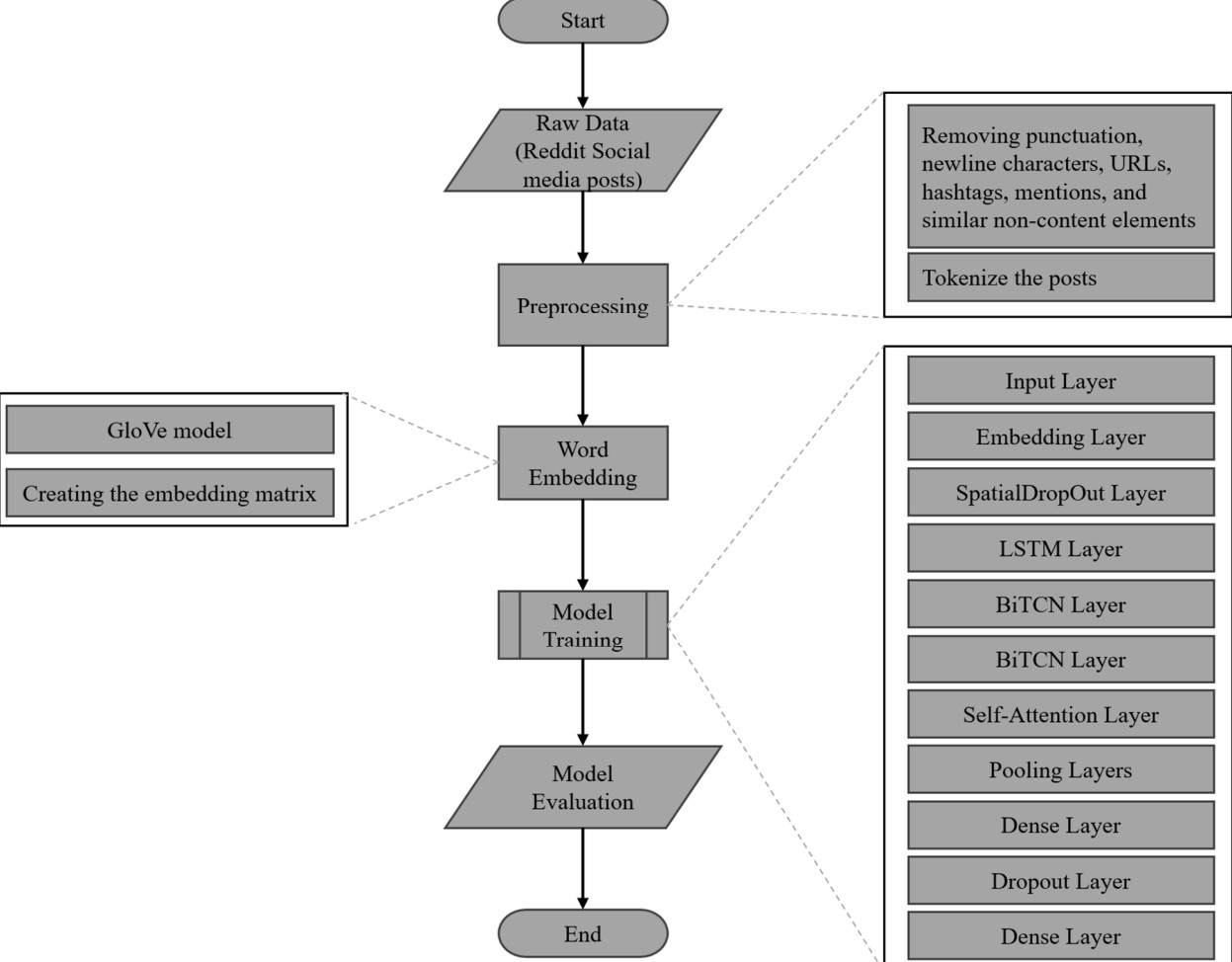

**Figure 1.** Architecture of the proposed model.

### 3.2. Word Embedding

In our research, we utilized advanced word embedding techniques, specifically the GloVe (Global Vectors for Word Representation) model [14], to develop a robust vector space representation of words. The GloVe model operates on the principles of unsupervised learning, generating word vectors through a comprehensive analysis of word co-occurrence statistics derived from a given corpus. This approach hinges on constructing a global matrix that captures the frequency of word pair occurrences within the corpus. Each non-zero entry in this matrix signifies the co-occurrence of words, serving as a foundational dataset for training the GloVe model.

A unique aspect of the GloVe method is its ability to reveal intriguing linear substructures within the word vector space, highlighting relationships and patterns in the data. The initial step in employing the GloVe model involves a thorough traversal of the corpus to accumulate the necessary statistical data for populating the co-occurrence matrix. While this process may demand significant computational resources, particularly for larger corpora, it is important to note that this is a one-time investment. Once the matrix is

populated and the model is trained, the resultant word embeddings can be used for various analytical purposes without the need to repeat this computationally intensive step [14].

In our research, we have incorporated the utilization of pre-trained lexical representations provided by the Stanford NLP Group. Specifically, we employed the extensive corpus variant, known as "Common Crawl (840 B tokens, 2.2 M vocab, cased, 300 d vectors)", which encompasses an impressive 840 billion tokens, a vocabulary of 2.2 million cased terms, and provides each term with a 300-dimensional vector. This corpus is publicly accessible on the Stanford NLP group's GloVe project page (https://nlp.stanford.edu/projects/glove/, accessed on 2 October 2023). Within our framework, the GloVe model's role was to furnish the initial embedding layer by mapping each token derived from our dataset into a 300-dimensional semantic space.

### 3.3. Proposed LSTM-Attention-BiTCN Model

#### 3.3.1. Long Short-Term Memory (LSTM)

In this research, we incorporated Long Short-Term Memory (LSTM) networks, a specialized form of Recurrent Neural Networks (RNNs) renowned for their ability to capture and preserve long-range dependencies within sequential data. Unlike traditional RNNs, LSTMs exhibit a heightened capacity to remember information over extended periods, thus mitigating the commonly encountered vanishing gradient issue often associated with standard RNNs [15]. The core functionality of LSTM networks resides in their unique architecture, characterized by three principal gates within each neuron: the forget gate, input gate, and output gate. These gates collectively govern the flow and modulation of information through the network. They execute this control by leveraging the hyperbolic tangent and sigmoid activation functions, which play a crucial role in updating and maintaining the cell state.

In the proposed model, we employed an LSTM layer comprising 100 units. This layer was tasked with receiving the vector representations processed by a preceding spatial dropout layer. The LSTM layer then outputs a sequence of values, which effectively encapsulate the encoded information from the input vectors, tailored for subsequent stages of the neural network model. This design choice was guided by the necessity to capture both the spatial and temporal dynamics inherent in the dataset under analysis.

#### 3.3.2. Bidirectional Temporal Convolutional Networks (BiTCNs)

In our study, we also delved into the application of Temporal Convolutional Networks (TCNs) [16], an architecture explicitly crafted for sequential data processing, which amalgamates the principles of causal convolutions and dilation. TCNs are distinguished by two principal features. Firstly, the causality inherent in their convolutional layers ensures that information from future time steps does not inadvertently influence earlier ones. Additionally, their architectural design enables the mapping of input sequences of any length to outputs of equivalent length.

The cornerstone of TCNs is the implementation of dilated causal convolutions, which are instrumental in expanding the receptive field without necessitating an increase in the number of layers. The operation of dilated convolution on a one-dimensional input sequence is mathematically represented in Equation (1), incorporating a filter $f = \{0, 1, \ldots, k-1\} \rightarrow R$, where $k$ is the filter size, $d$ is the dilation factor, and $s - d \cdot i$ indicates the prior data points.

$$F(s) = \sum_{i=0}^{k-1} f(i) \cdot x_{s-d \cdot i} \tag{1}$$

TCNs feature a residual block structure formed by stacking two convolutional layers. If there is a discrepancy between the lengths of the input and output, an additional one-dimensional convolution is integrated into the residual block to ensure output vectors match the input length. The construction of a TCN involves stacking a requisite number

of these residual blocks to address the specific requirements of the problem at hand. Figures 2 and 3 illustrate the fundamental architecture of a TCN, highlighting both the dilated causal convolution and the corresponding residual block. Each block comprises two dilated causal convolution layers that are enhanced by non-linear activation functions and weight normalization [16].

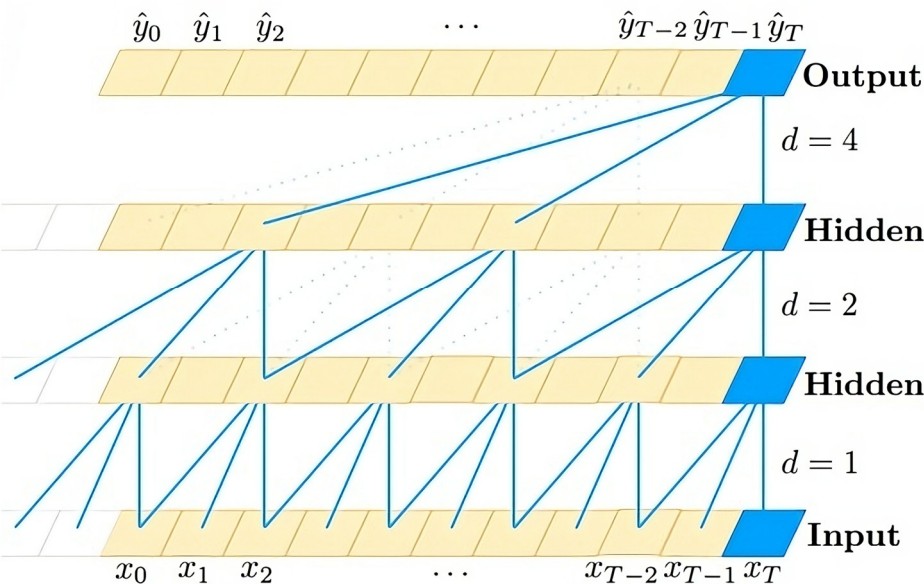

**Figure 2.** Architecture of a dilated causal convolution.

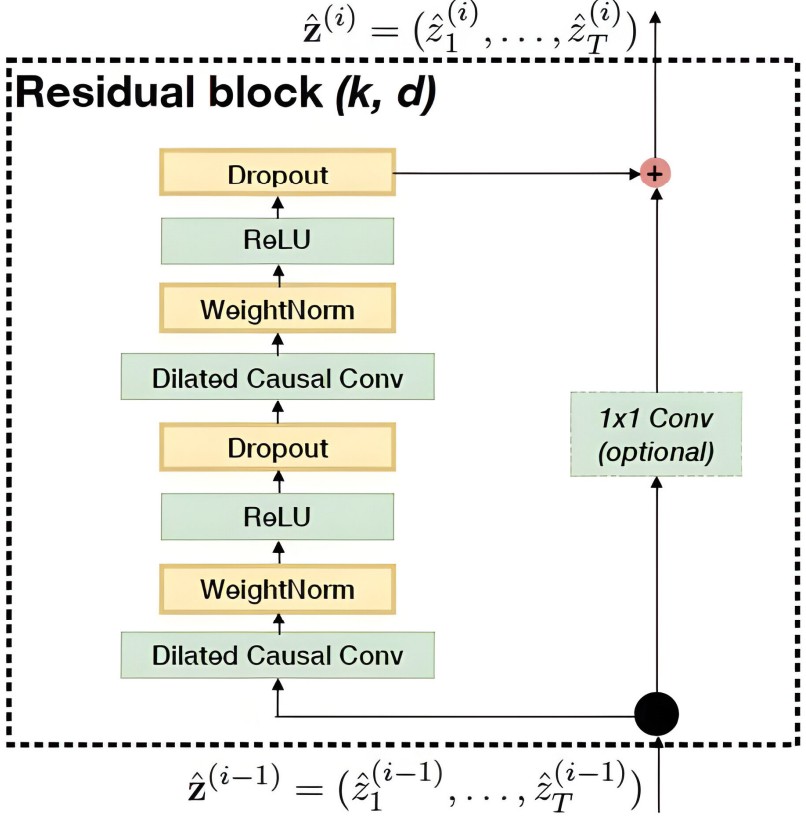

**Figure 3.** Architecture of the residual block.

Previous studies [16,17] have demonstrated TCNs' superiority over LSTM models in certain natural language processing (NLP) tasks. In our proposed model, we integrate two successive hidden layers of Bidirectional Temporal Convolutional Networks (BiTCNs). This choice is predicated on the ability of BiTCNs to thoroughly assimilate contextual information from the embedded vectors derived through word embedding techniques [18]. By processing both forward and backward sequences of words within sentences, BiTCNs adeptly capture the intricate dependencies and relationships between words, thereby enriching the model's interpretative capabilities.

### 3.3.3. Attention Layer

The concept of the attention mechanism was initially introduced to enhance the performance of machine translation tasks [19], and it serves as a pivotal component in our LSTM-Attention-BiTCN model. This mechanism operates by emulating human attention processes, integrating a deep learning layer designed to emphasize the more salient pieces of information through data weighting. An attention layer can be essentially conceptualized as computing the weighted average of the input data, thereby reducing its dimensionality or rank.

In our model, we strategically positioned a self-attention layer subsequent to the TCN layers. The role of this self-attention layer was to meticulously extract and accentuate the essential elements within the posts fed into the model. This is crucial for ensuring that the model focuses on the most informative parts of the text, thereby enhancing the accuracy and relevance of its output. The computational process of the self-attention layer in our model is encapsulated in Equations (2)–(4):

$$U = \tan h(XW_t + b_t) \tag{2}$$

$$a = softmax(U) \tag{3}$$

$$output = X \times U \tag{4}$$

### 3.3.4. Dropout Layers

In our research, we integrated the dropout layer strategy, an essential technique for mitigating the risk of overfitting within machine learning models. Overfitting is particularly prevalent in scenarios where large neural networks are trained on relatively small datasets. The dropout layer addresses this issue by randomly nullifying a proportion of the input units during each training iteration. This process is controlled by a predetermined frequency, known as the dropout rate, which lies within the range of [0, 1]. Concurrently, the remaining active units are scaled up by a factor of $1/(1 - rate)$, ensuring the total input signal's intensity remains consistent.

In the architecture of our proposed LSTM-Attention-BiTCN model, we employed a spatial dropout layer, a variant of the standard dropout layer, which is particularly advantageous in the context of convolutional networks. This spatial dropout layer was strategically positioned immediately following the embedding layer. Its primary function was to randomly deactivate the entire feature maps in the convolutional layers, as opposed to individual neurons, thereby enhancing the model's generalization capabilities.

Additionally, we incorporated another dropout layer just before the final dense layer of the model. Both dropout layers, the spatial dropout layer and the standard dropout layer, were configured with a dropout rate of 0.1. This rate was carefully chosen to strike a balance between preventing overfitting and maintaining sufficient information flow through the network. By employing these dropout layers at these critical junctures in the model, we aim to enhance the robustness and generalizability of the LSTM-Attention-BiTCN model, thereby improving its performance in accurately detecting and analyzing patterns indicative of suicidality in social media posts.

## 4. Experimental Results

*4.1. Dataset and Exploratory Analysis*

4.1.1. Dataset

We utilized a publicly accessible dataset sourced from Reddit, which was obtained from the Kaggle platform [20]. This dataset is comprehensive, encompassing a total of 232,074 posts in English, which were collected from the SuicideWatch subreddit. The term "SuicideWatch" is typically associated with a vigilant monitoring process aimed at preventing suicide attempts. This procedure is commonly implemented in environments such as jails, hospitals, mental health facilities, and military bases. It is specifically designated for individuals who are perceived to exhibit warning signs of suicide, suggesting a heightened risk of self-harm. The subreddit 'SuicideWatch' is a digital manifestation of this concept, where individuals potentially exhibiting suicidal tendencies or thoughts share their experiences and seek support. The timeframe for this data collection spans from 16 December 2008 to 2 January 2021. The dataset is meticulously balanced, consisting of 116,037 posts identified as suicidal and an equal number of posts classified as non-suicidal. The average length of each post in the dataset is approximately 430 words. Figure 4 illustrates the dataset, which consists of two columns: "text", which provides the content of user postings, and "class", which contains the label associated with the text.

| A text | A class |
| --- | --- |
| Ex Wife Threatening SuicideRecently I left my wife for good because she has cheated on me twice and ... | suicide |
| Am I weird I don't get affected by compliments if it's coming from someone I know irl but I feel rea... | non-suicide |
| Finally 2020 is almost over... So I can never hear "2020 has been a bad year" ever again. I swear to... | non-suicide |
| i need helpjust help me im crying so hard | suicide |
| I'm so lostHello, my name is Adam (16) and I've been struggling for years and I'm afraid. Through th... | suicide |
| Honetly idkI dont know what im even doing here. I just feel like there is nothing and nowhere for me... | suicide |

**Figure 4.** Dataset example.

To evaluate the performance of both the proposed model and the baseline models, the dataset was stratified–split into three distinct subsets: training, testing, and validation, adhering to a proportional division of 70:20:10, respectively. The training subset, consisting of 162,452 samples, was utilized to train the models. The testing subset, which included 46,415 samples, served the purpose of evaluating the models' performance and generalization capabilities on unseen data. Lastly, the validation subset, containing 23,207 samples, was employed as an intermediary check during the model training process, aiding in the optimization of hyperparameters and helping to prevent overfitting.

### 4.1.2. Word Cloud Analysis

In this section, we employed word cloud analysis to delve into the linguistic and statistical attributes of suicide-related communications, with a particular focus on differentiating these from regular social media posts. Figure 5 revealed that words such as "like", "want", "feel", and "know" frequently appear in the content of suicidal notes. However, it is noteworthy that these words are predominantly used in their negated forms, including phrases like "don't like", "don't want", "don't feel", and "don't know". The recurrent usage of these negations points toward a pattern of reluctance or denial in expressing positive sentiments or desires. The predominance of such negations may be indicative of underlying depressive states or suicidal ideation. This linguistic tendency reflects a sense of dissatisfaction, helplessness, or lack of control over personal emotions and circumstances.

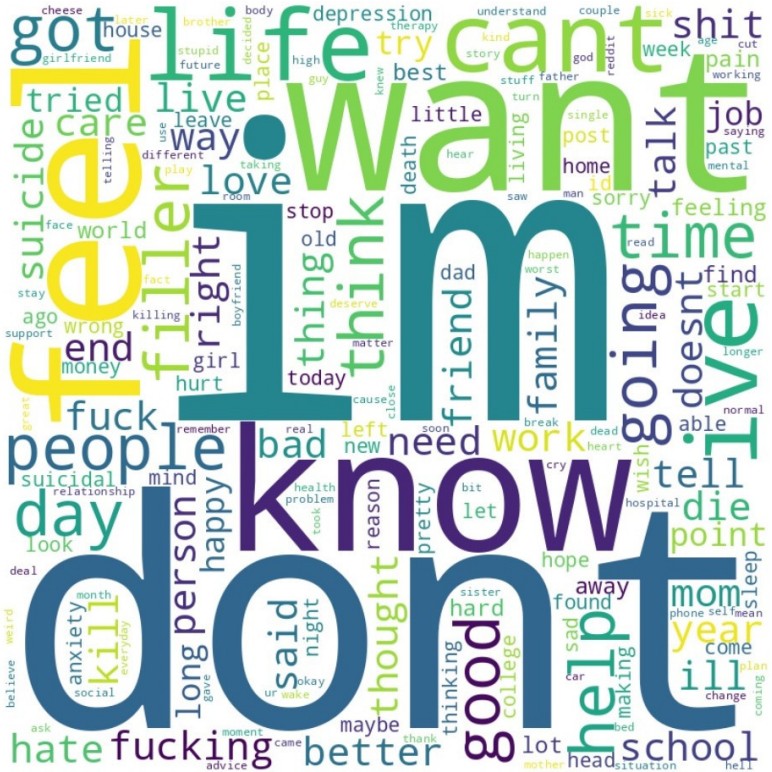

**Figure 5.** Word cloud analysis.

### 4.1.3. N-Gram Analysis

In our analysis results, as depicted in Figure 6, we observed the prevalence of certain bi-grams in the dataset, offering a deeper insight into the collective mindset of the authors of these posts. The bi-gram "don't know" emerged as the most frequent, appearing 47,675 times. This was closely followed by "feel like" and "don't want", with occurrences of 44,661 and 38,755 times, respectively. Additionally, the analysis highlighted the recurrence of expressions like "I'm going", "fuck fuck", and "like I'm", each appearing over 12,000 times. Furthermore, expressions such as "want die", "don't think", and "I'm tired"

were notably recurrent. The phrase "want die" is particularly alarming, as it directly conveys a disturbing inclination toward ending one's life. The frequent appearance of "suicidal thoughts", which was mentioned 5910 times, further accentuates the serious nature of the communications within this dataset.

|     | Frequency | Bigram |
| --- | --- | --- |
| 1 | 47,675 | dont know |
| 2 | 44,661 | feel like |
| 3 | 38,755 | dont want |
| 4 | 16,112 | im going |
| 5 | 12,839 | fuck fuck |
| 6 | 12,529 | like im |
| 7 | 12,268 | want die |
| 8 | 10,967 | dont think |
| 9 | 10,881 | im tired |
| 10 | 10,249 | dont even |
| 11 | 10,066 | know im |
| 12 | 8911 | every day |
| 13 | 7755 | get better |
| 14 | 7538 | even though |
| 15 | 7259 | high school |
| 16 | 7160 | feels like |
| 17 | 7057 | think im |
| 18 | 7055 | cant even |
| 19 | 7002 | im scared |
| 20 | 6525 | dont care |
| 21 | 6490 | ive never |
| 22 | 6330 | dont feel |
| 23 | 6279 | im sorry |
| 24 | 6134 | im gonna |
| 25 | 6082 | dont really |
| 26 | 6080 | im sure |
| 27 | 5912 | best friend |
| 28 | 5910 | suicidal thoughts |
| 29 | 5654 | years ago |
| 30 | 5497 | ive tried |

**Figure 6.** Bi-grams analysis.

The aforementioned phrases illustrate that individuals experiencing suicidal ideation often communicate their melancholy and negative emotions and sentiments in their online posts, both implicitly and explicitly. Our analysis showed that these expressions often involve negations of verbs like "want" or "feel". This linguistic trend suggests a prevailing reluctance among these individuals to continue enduring their suffering. Recognizing these expressions in social media posts can be pivotal in the early detection of suicidal tendencies, potentially averting the exacerbation of their mental health conditions.

Intriguingly, the analysis also unearthed the presence of seemingly positive terms in these posts, such as "get better", "high school", and "best friends". At first glance, these terms may appear contradictory to the overarching theme of negativity and despair. However, it is crucial to understand that the mere frequency of such ostensibly positive terms does not negate the underlying distress. Their inclusion in discussions about suicidal ideation often highlights the contrast between what is desired or remembered fondly and the current state of suffering, thereby emphasizing the depth of the individual's struggle.

Therefore, it is imperative to approach the analysis of such posts with a nuanced perspective. Focusing solely on specific terms or phrases may not provide an accurate representation of the individual's mental state. It is more beneficial to monitor the changes in mood over time and consider the full context of their communication. The identification of n-grams commonly utilized by individuals experiencing suicidal ideation can contribute to the timely detection of this complex mental health concern.

### 4.1.4. Topics Analysis

In our continued exploration of textual data for insights into mental health discourse, we turned our analysis to topic modeling using the sklearn library's Non-negative Matrix Factorization (NMF) algorithm. As illustrated in Figure 7, the topic detection model revealed several themes that shed light on the underlying sentiments and concerns expressed by social media users. Upon closer inspection of the model's output, we discerned a pattern suggestive of distress and existential dread. The first topic, with phrases like "don't know", "feel like", and "like I'm", potentially captures the ambivalence and introspection characteristic of individuals grappling with deep-seated uncertainties. The second topic, encompassing terms such as "I'm bored", "anyone wanna", and "wanna talk", may reflect a yearning for interaction or a plea for companionship, which is often lacking in individuals contemplating suicide. Moreover, the third topic, containing "among us", "guys girls welcome", and "commitment guys girls", hints at a search for community or belonging, possibly indicating the social dimension of the users' experiences. Finally, the fourth topic starkly presents expressions such as "don't want", "want die", and "want live", directly pointing to a conflict between the will to live and the inclination toward self-harm.

## Topic Detection

```python
from sklearn.feature_extraction.text import TfidfVectorizer
from sklearn.decomposition import NMF
from sklearn.pipeline import make_pipeline
tfidf_vectorizer = TfidfVectorizer(stop_words=stopwords, ngram_range=(2,3))
nmf = NMF(n_components=5)
pipe = make_pipeline(tfidf_vectorizer, nmf)
pipe.fit(df['Text'])

print_top_words(nmf, tfidf_vectorizer.get_feature_names(), n_top_words=3)
```

```
Results:
Topic #1: dont know, feel like, like im
Topic #2: im bored, anyone wanna, wanna talk
Topic #3: among us, guys girls welcome, commitment guys girls
Topic #4: dont want, want die, want live
```

**Figure 7.** Topics analysis.

These topics, in aggregate, suggest not only a prevalence of suicidal ideation and depressive thoughts but also illuminate the multifaceted nature of the conversation around mental health. The presence of terms like "want die" alongside "want live" within the same topic underscores the complexity of suicidal ideation, which often includes ambivalent feelings about life and death. This nuanced understanding of the topics prevalent in social media discussions provides valuable context for mental health professionals. It may inform

the development of targeted interventions and support systems, potentially enabling the early detection and prevention of suicide among individuals with mental health challenges.

*4.2. Evaluation Metrics and Experimental Setup*

In our research, the primary objective of the LSTM-Attention-BiTCN model, and indeed the baseline methods, is to accurately classify social media posts in terms of suicidal ideation. To assess the efficacy of these models, we employ a suite of metrics commonly used in classification tasks: accuracy, precision, recall, and the F1-score, as represented in Equations (5)–(8).

$$Accuracy = \frac{True_{possitive} + True_{negative}}{Total\ predictions} \tag{5}$$

$$Precision = \frac{True_{possitive}}{True_{positive} + False_{positive}} \tag{6}$$

$$Recall = \frac{True_{possitive}}{True_{positive} + False_{negative}} \tag{7}$$

$$F1 - score = 2 \times \frac{Recall \times Precision}{Recall + Precision} \tag{8}$$

For the experimental setup of our study, we conducted the implementation and evaluation of all baseline methods, encompassing traditional machine learning models, advanced deep learning techniques, and attention-based ensemble models by using the Google Colab Notebook service with Tesla A100 GPU and 84 GB RAM. Our development and testing were carried out using Python version 3.10. For the construction and training of deep learning models, we employed Keras, a powerful and user-friendly open-source library. Additionally, we leveraged the capabilities of the Natural Language Toolkit (NLTK), a comprehensive Python library designed specifically for NLP tasks. A key component of our proposed model included TCN-based layers implemented using the Keras Temporal Convolutional Network library. The LSTM-Attention-BiTCN model was trained utilizing the "adam" optimizer with a default learning rate of 0.001 and a "binary cross entropy" loss function for a total of 30 epochs and 64 batches. To prevent model overfitting and accelerate the training speed, we applied two callback functions: EarlyStopping and ReduceLROnPlateau. The EarlyStopping callback intervenes to halt the training when there is no improvement in validation accuracy for a specified patience of 10 epochs, thus ensuring the model retains the best weights at the point of maximum validation accuracy. Concurrently, the ReduceLROnPlateau callback dynamically adjusts the learning rate when the validation loss fails to decrease over a patience period of five epochs, with the learning rate being reduced by a factor of 0.1. Table 1 presents the quantities of units and associated configurations, along with the model's architectural design.

**Table 1.** Experimental setup of the proposed LSTM-Attention-BiTCN.

| Layer | Parameters | Values |
|---|---|---|
| Embedding | Embedding dimension | 300 |
| Spatial Dropout | Dropout rate | 0.1 |
| LSTM | Units, activation function | 100, tanh |
| Bidirectional TCN (1) | Units, activation function, dilations, dropout rate | 128, ReLu, [1, 2, 4], 0.1 |
| Bidirectional TCN (2) | Units, activation function, dilations, dropout rate | 64, ReLu, [1, 2, 4], 0.1 |
| Dense (1) | Units, activation function | 16, ReLu |
| Dropout | Dropout rate | 0.1 |
| Dense (2) | Units, activation function | 1, sigmoid |

### 4.3. Classification Results

4.3.1. Training and Validation Performance of the LSTM-Attention-BiTCN Model

The proposed LSTM-Attention-BiTCN model underwent an optimization process spanning 30 epochs with a set learning rate of 0.001. The accuracy trends depicted in Figure 8 illustrate that the model's performance improved steadily over the course of 30 epochs. The graph indicates a positive trajectory in both training and validation accuracies, with the training accuracy initiating at approximately 0.91 and ascending to just over 0.95, while the validation accuracy began near 0.89 and rose to about 0.93. This convergence of training and validation lines suggests that the model generalizes well and is not overfitting.

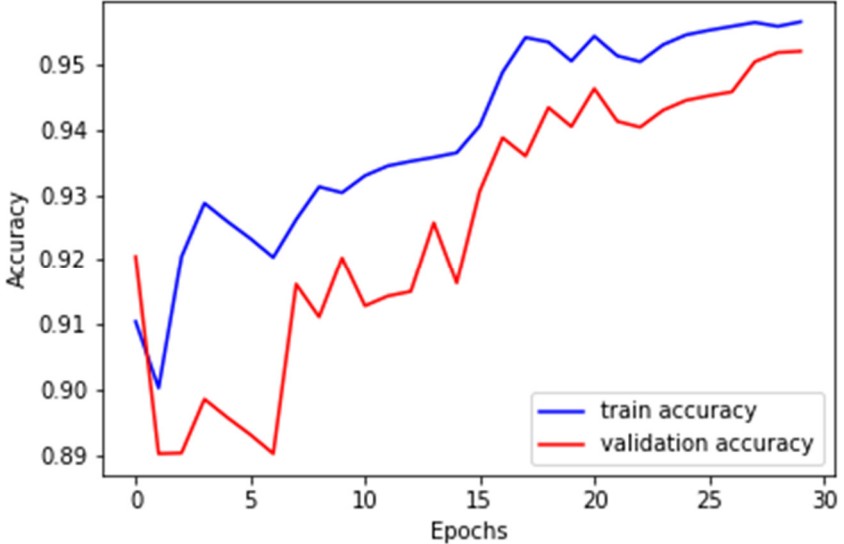

**Figure 8.** Training and validation accuracy.

Figure 9 portrays the trajectory of the training and validation losses throughout the model's optimization phase. Both metrics exhibit a downward trend, with the training loss initiating at higher levels and concluding just above 0.1 after 30 epochs. Conversely, the validation loss starts slightly lower and converges toward a nearly similar value by the end of the training period. This parallel reduction suggests that the LSTM-Attention-BiTCN model, assessed via binary cross-entropy, is learning effectively without overfitting.

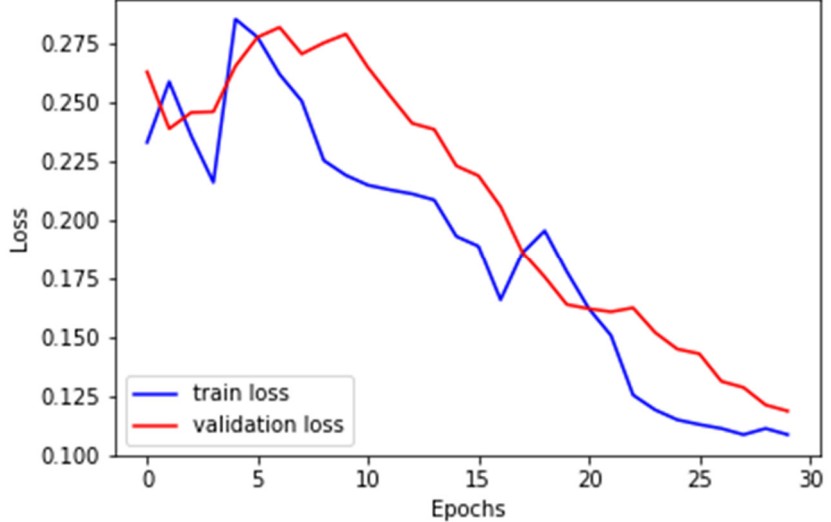

**Figure 9.** Training and validation loss.

4.3.2. Traditional Machine Learning Baselines

In this part of our study, we undertook a comparative analysis of the LSTM-Attention-BiTCN model against a range of conventional machine learning algorithms, including K-Nearest Neighbor [21], Random Forest [22], Decision Tree [23], Gradient Boost [24], XGBoost [25], and Logistic Regression [26]. The primary objective was to assess the efficacy of these models in differentiating between suicide-related text and regular posts on Reddit.

For this comparison, all models utilized the GloVe word embedding technique, processing inputs as embedded vectors corresponding to the words in the dataset. However, when training traditional machine learning algorithms, the embedded vectors were generated by taking the average of the GloVe embedding vectors of all words contained within the document. This approach ensures compatibility with the traditional machine learning models. The training of these models was conducted using their default hyperparameters. The results, as detailed in Table 2, provide a comparative overview of the performance on the testing subset of six traditional machine learning models against our LSTM-Attention-BiTCN model.

**Table 2.** The comparison performance of machine learning baseline models and the proposed model.

| Baseline Models | Accuracy | Precision | Recall | F1-Score |
|---|---|---|---|---|
| K-Nearest Neighbor | 0.7195 | 0.7196 | 0.7195 | 0.7195 |
| Random Forest | 0.8037 | 0.8037 | 0.8037 | 0.8037 |
| Decision Tree | 0.7687 | 0.7688 | 0.7687 | 0.7687 |
| Gradient Boost | 0.8005 | 0.8005 | 0.8005 | 0.8005 |
| XGBoost | 0.8396 | 0.8401 | 0.8396 | 0.8396 |
| Logistic Regression | 0.7257 | 0.7351 | 0.7257 | 0.7228 |
| LSTM-Attention-BiTCN | 0.9405 | 0.9385 | 0.9424 | 0.9405 |

Among the traditional models, XGBoost exhibited superior performance, achieving an accuracy of 0.8396, a precision of 0.8401, a recall of 0.8396, and an F1-score of 0.8396. Following closely, the Random Forest and Gradient Boost models demonstrated competitive results, with all performance metrics' scores of 0.8037 and 0.8005, respectively. The Decision Tree, Logistic Regression, and K-Nearest Neighbor models were ranked in that order.

However, when compared to these traditional machine learning models, our LSTM-Attention-BiTCN model showcased a significantly higher performance in accurately identifying suicidality in social media text. The LSTM-Attention-BiTCN model achieved an accuracy of 0.9405, a precision of 0.9385, a recall of 0.9424, and an F1-score of 0.9405. These results underline the effectiveness and robustness of the LSTM-Attention-BiTCN model in the context of suicide ideation detection on social media platforms, surpassing the capabilities of the various traditional classification models examined. This comparison highlights the advanced potential of LSTM-Attention-BiTCN in addressing complex classification challenges within the realm of social media content analysis.

4.3.3. Deep Learning Baselines

In this section, we delve into the performance of various deep learning models, specifically focusing on LSTM and CNN architectures, as well as a hybrid model that combines LSTM and CNN (LSTM-CNN). These models are evaluated and compared to our proposed LSTM-Attention-BiTCN model. The configuration details for each of these deep learning models are as follows:

- LSTM-CNN Ensemble Model: This model integrates the features of both LSTM and CNN. It comprises an input layer, an embedding layer, and a single LSTM layer with 100 units, followed by a CNN layer equipped with three filters, a kernel size of eight, and employing a ReLu activation function. The model structure includes a dropout layer after the initial input layer, with a dropout rate of 0.1, and a flattening layer preceding the final dense layer. The dense layer, with a single unit and a "sigmoid" activation function, is responsible for the final classification task. Training of this model

utilized the "adam" optimizer at a default rate of 0.001 and binary cross-entropy loss, spanning up to 30 epochs and batch size of 64. Similar to the LSTM-Attention-BiTCN model, EarlyStopping and ReduceLROnPlateau callbacks were applied. The model training concluded early at epoch 24 with a learning rate of $1 \times 10^{-5}$.

- LSTM Model: This model begins with an input layer followed by an embedding layer and a dropout layer with a dropout rate of 0.1. A single LSTM layer with 100 units is situated immediately after the dropout layer. A flattening layer follows before culminating in the final dense layer, which mirrors the settings of the LSTM-CNN model designed for classification. The training process also utilized the "adam" optimizer with a standard rate of 0.001 and binary cross-entropy loss, spanning up to 30 epochs and a batch size of 64. Incorporating EarlyStopping and ReduceLROnPlateau callbacks, the training early stopped at epoch 21 with a final learning rate of $1 \times 10^{-4}$.

- Bidirectional LSTM (BiLSTM) Model: The architecture begins with an input layer, which is immediately followed by an embedding layer and a dropout layer set at a rate of 0.1. Next, a Bidirectional Long Short-Term Memory (BiLSTM) layer with 100 units is employed, allowing the model to process data in both forward and backward directions for a more comprehensive understanding. A flattening layer is then applied, ensuring that the output is structured appropriately for the final stage. The structure concludes with a dense layer, similar to the LSTM model. With the training settings mirroring those of the LSTM model, the BiLSTM model was stopped early at epoch 22 and had a final learning rate of $1 \times 10^{-5}$.

- CNN Model: In the CNN setup, the input layer is followed by an embedding layer and a dropout layer with an identical dropout rate as the other models. The key feature of this model is a one-dimensional convolution layer configured with a filter size of three and a kernel size of eight and utilizing a ReLu activation function. The architecture concludes with a flattening layer, leading into a dense layer that performs the classification, similar to the other models mentioned. The training parameters of this model included the "adam" optimizer at a default 0.001 rate and a binary cross-entropy loss over 30 epochs with a batch size of 64. The CNN model, implementing EarlyStopping and ReduceLROnPlateau callbacks, halted training early at epoch 24 with a learning rate of $1 \times 10^{-6}$".

Table 3 presents a comprehensive comparison of various deep learning models on the testing subset, highlighting their performance in the task of classifying suicidal ideation in text content. The baseline deep learning models, including LSTM-CNN, LSTM, and BiLSTM, showed moderate performance with scores ranging from 0.7534 to 0.7752. However, the CNN model showed lower performance in accuracy and precision compared to the others. When compared to these models, our proposed LSTM-Attention-BiTCN model demonstrated exceptional performance, markedly outperforming the deep learning baselines. It achieved scores above 0.91 across all metrics. This superior performance underscores the effectiveness of the LSTM-Attention-BiTCN model in accurately detecting and classifying suicidal content in social media posts, affirming its potential as a valuable tool in mental health monitoring and intervention efforts.

**Table 3.** The comparison performance of deep learning baseline models and the proposed model.

| Baseline Models | Accuracy | Precision | Recall | F1-Score |
| --- | --- | --- | --- | --- |
| LSTM-CNN | 0.7623 | 0.7546 | 0.7752 | 0.7647 |
| LSTM | 0.7594 | 0.7612 | 0.7534 | 0.7573 |
| BiLSTM | 0.7610 | 0.7623 | 0.7561 | 0.7592 |
| CNN | 0.6843 | 0.6344 | 0.8649 | 0.7319 |
| LSTM-Attention-BiTCN | 0.9405 | 0.9385 | 0.9424 | 0.9405 |

## 5. Conclusions and Future Research

In this research, we explored the potential of leveraging social media content, specifically posts from Reddit, to detect suicidal ideation using advanced deep learning techniques.

We developed a novel ensemble learning approach named LSTM-Attention-BiTCN, which integrates LSTM and BiTCN models with self-attention to identify signs of suicidality in social media posts. The primary aim of this model is to aid healthcare professionals in recognizing suicidal tendencies among social media users, thereby contributing to efforts to reduce suicide rates. The LSTM-Attention-BiTCN model offers dual utility: it can be employed by social network services to gauge the likelihood of suicidal thoughts among their users, and it can also serve healthcare providers as a tool for monitoring the mental health of their patients. Based on the experimental evaluations of our model, we observed that it exhibits a high degree of accuracy in detecting suicidal ideation. It outperformed both traditional and contemporary machine learning and deep learning models. The LSTM-Attention-BiTCN model demonstrated superior performance, with an accuracy of 0.9405, a precision of 0.9385, a recall of 0.9424, and an F1-score of 0.9405.

Looking ahead, we plan to explore a variety of deep learning techniques and their combinations to discover even more effective ensemble models. A crucial aspect of future research will be addressing out-of-vocabulary (OOV) issues, which are significant in social network post analysis. We plan to implement strategies such as subword tokenization or the FastText embedding technique to enhance model robustness against rare or unseen words. We also aim to examine different attention mechanisms to assess their impact on model performance. Future research could expand upon this work by incorporating diverse word embedding techniques and integrating various behavioral and social network profile features into the model. We also suggest analyzing multiple social media posts over different time intervals to better capture changes in behavior and mental condition. Additionally, other relevant factors such as location, application usage time, and social connections could be included to enhance the model's comprehensiveness and accuracy in detecting suicidal ideation. Finally, we propose investigating the integration of Large Language Models (LLMs) into our LSTM-Attention-BiTCN framework to improve the contextual analysis of social media posts. This integration could offer a more holistic understanding of user sentiment and behavioral patterns by leveraging the LLM's ability to analyze large datasets and generate human-like responses.

**Author Contributions:** Conceptualization, H.-S.C. and J.Y.; methodology, H.-S.C. and J.Y.; software, H.-S.C.; validation, H.-S.C. and J.Y.; formal analysis, H.-S.C.; investigation, H.-S.C. and J.Y.; writing—original draft preparation, H.-S.C.; writing—review and editing, H.-S.C.; visualization, H.-S.C.; supervision, J.Y. All authors have read and agreed to the published version of the manuscript.

**Funding:** This research received no external funding.

**Institutional Review Board Statement:** Not applicable.

**Informed Consent Statement:** Not applicable.

**Data Availability Statement:** The data presented in this study are available here: https://www.kaggle.com/datasets/nikhileswarkomati/suicide-watch, accessed on 28 August 2023.

**Conflicts of Interest:** The authors declare no conflicts of interest.

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
