# Peer review of "RETRACTED: Innovative Use of Self-Attention-Based Ensemble Deep Learning for Suicide Risk Detection in Social Media Posts"

_applsci, doi:10.3390/app14020893_

Round 1

Reviewer 1 Report

Comments and Suggestions for Authors

This paper addresses the critical concern of suicidal ideation in mental health, emphasizing the importance of early detection for timely support. The paper introduces an innovative ensemble learning method called LSTM-Attention-BiTCN. This model demonstrated superior performance in detecting signs of suicidality in social media posts, achieving an accuracy of 93.55%, precision of 94.65%, recall of 92.27%, and an F1-score of 93.44% compared to baseline models. The proposed model is suggested to be a valuable tool for healthcare professionals. However, I propose some minor changes:

- At the end of the introduction, i would be interesting adding a paragraph outlining the paper's structure, including all the sections and their content.

- Even though the dataset is publicly available, enriching the paper with specific examples would provide clarity for the reader, allowing them to gain insight without the need to search for it on Kaggle.

-It would be interesting to mention in future work the impact of the popular LLM in this kind of researches

-It would be beneficial to include some charts illustrating the training progress to enhance understanding.

Comments on the Quality of English Language

Minor editing of English language require.

Author Response

We would like to thank the reviewers for their careful and thorough reading of the manuscript, as well as their thoughtful comments and constructive suggestions, which have helped improve the quality of the manuscript.

Reviewer 2 Report

Comments and Suggestions for Authors

Dear authors, 

The presented study is interesting, however it contains currently some serious issues mainly in the comparison with other deep learning methods section. The other methods that were used the input got a 50% dropout that was not present in the BiTCN architecture which clearly causes unfair comparison between these structures.

This part of the paper must be completely recalculated and if needed, rewritten.

Please find my remarks below.

The language of the dataset should be mentioned.

Definition of Accuracy, Precision recall and F1 is not needed, since these are basic values.

"Previous study [17] demonstrated TCNs' superiority over LSTM models in certain natural language processing (NLP) tasks." Well, citation [17] uses the TCN architecture indeed, but not on an NLP task since that paper is about protein secondary structure prediction. Please fix this in order to make the statement true.

Some info about the length of the data (how many characters/words on average) would be beneficial.

Describing the BiTCN structure is important, however in my opinion providing the equations for LSTM is not necessary sine that is a well-known structure.

In L. 411-415 you write about how the documents were vectorized for the traditional ML algorithms. However, it is not mentioned how the vectors were used to create a document embeddig. did you apply averaging the word embeddings or other technique? Please specify.

L- 416-430. You mention results but it is not clear how the data was split for testing. Please provide this information.

L-438-457. The input dropout in each model is 0.5 that is 5 time more than the dropout used in the BiTCN architecture. Moreover, the BiTCN architecture's input did not have any dropout but only on later layers. This setting is unfair and leads to wrong results in terms of the comparison. This is a serious error. I suggest keeping as much variables the same as possible compared to the BiTCN structure. Otherwise the comparison is not fair and scientific.

In my humble opinion the BiLSTM structure should be also tested, by e.g. concatenating the memory vectors from both directions. Using that coud provide a firmer baseline.

Comments on the Quality of English Language

Good.

Author Response

(The authors gave the same response as above.)

Reviewer 3 Report

Comments and Suggestions for Authors

The authors presented an interesting paper on identifying the suicidal risk. This is an important area of research to reduce the suicide intention. The authors achieve significantly high performance for their proposed model based on LSTM and BiTCN. The authors use Reddit posts for training and testing the proposed model.

The manuscript is well written. The authors conducted different experiments to establish the performance of the proposed model. The results achieved are satisfactory.

Major revisions:

1.     In Figure 1, Model training includes two Dense layers. What is the difference between these layers?

2.     Is the proposed model applicable to find the suicidal risks from other social media text (ex. Twiter)?  

Author Response

(The authors gave the same response as above.)

Round 2

Reviewer 2 Report

Comments and Suggestions for Authors

Dear Authors,

The paper have improved a lot, I am particularly happy about the involvement of BiLSTM architecture in the paper.

However, I have found some important points that must be still added. It may seem a lot but the majority can be added swiftly.

The train-dev-test split describing part imrpoved a lot, however one mindor detail is missing, namely, how the data was selected to the 70-10-20 % parts? Randomly? Or did you apply stratified sampling? Please add this information.

L. 427. You mention that the training lasted 30 epochs, but I did not find any information about the learning rate used during training. Was it fixed, or changed during the training? What were the settings? Please add this information also to the other deep learning methods.

Fig 8 and 9. Please add the axis titles, accuracy [%] and epochs.

The addition of Lines 179-187, section 3.2 is good, however I still miss the information I meant, which is still not clear for me in the text that in case of the traditional ML methods what was the method to create the embedding representations for the documents. GloVe provides a vector for each word that was present in your corpus. In order to get a vector representation were these vectors averaged, or did you apply other pooling techniques, maybe you used the dense vector logits from your proposed neural structure? I hope my question is clear now. Please specify this in the text (Section 4.3.2). 

It is clear that in case of the deep learning based solutions you applied an embedding matrix.

Also, it would be important to mention how you handled out of vocabulary words. Did you ignore them, of maybe used zero vectors maybe other solution?

ReLu was written with incosistent spelling throughout the text, sometimes ReLu, sometimes Relu, properly ReLU.

Section 4.3.3 describes the build ups of the different deep learning models but it is not clear how these models were trained, every settings were the same regardless of the possible overfitting or maybe early stopping was applied? Epochs, learning rate, checking for overfitting. So the details of the training of these models needs more clarification. Otherwise the comparison may be unfair again, since as Fig. 8 and 9 show the proposed new architecture was trained correctly but training e.g. the BiLSTM for the same epochs and learning rate could result in overfitting, hence bad results.

Comments on the Quality of English Language

Good.

Author Response

We would like to thank the reviewer for your careful and thorough reading of the manuscript, as well as your thoughtful comments and constructive suggestions, which have helped improve the quality of our manuscript.
